# Selection and Validation of Reference Genes for Gene Expression Studies in an Equine Adipose-Derived Mesenchymal Stem Cell Differentiation Model by Proteome Analysis and Reverse-Transcriptase Quantitative Real-Time PCR

**DOI:** 10.3390/genes14030673

**Published:** 2023-03-08

**Authors:** Angela L. Riveroll, Sabrina Skyba-Lewin, K. Devon Lynn, Glady’s Mubyeyi, Ahmad Abd-El-Aziz, Frederick S. T. Kibenge, Molly J. T. Kibenge, Alejandro M. Cohen, Blanca Esparza-Gonsalez, Laurie McDuffee, William J. Montelpare

**Affiliations:** 1Department of Applied Human Sciences, University of Prince Edward Island, 550 University Avenue, Charlottetown, PE C1A 4P3, Canada; 2Diagnostic Services and Adjunct, Department of Pathology and Microbiology, Atlantic Veterinary College, University of Prince Edward Island, Charlottetown, PE C1A 4P3, Canada; 3Department of Biology, University of Prince Edward Island, Charlottetown, PE C1A 4P3, Canada; 4Department of Pathology and Microbiology, Atlantic Veterinary College, University of Prince Edward Island, Charlottetown, PE C1A 4P3, Canada; 5Biological Mass Spectrometry Core Facility, Dalhousie University, Halifax, NS B3H 4R2, Canada; 6Department of Health Management, Atlantic Veterinary College, University of Prince Edward Island, Charlottetown, PE C1A 4P3, Canada

**Keywords:** equine adipose-derived mesenchymal stem cell differentiation, reference genes, gene expression, protein expression, *FABP5*, *RUNX2*, *PPP6R1*, *CCDC97*, *ACTB*, *EPHA2*

## Abstract

Adipose-derived stem cells (ADSCs) are used in tissue regeneration therapies. The objective of this study is to identify stable reference genes (RGs) for use in gene expression studies in a characterized equine adipose-derived mesenchymal stem cell (EADMSC) differentiation model. ADSCs were differentiated into adipocytes (ADs) or osteoblasts (OBs), and the proteomes from these cells were analyzed by liquid chromatography tandem mass spectrometry. Proteins that were stably expressed in all three cells types were identified, and the mRNA expression stabilities for their corresponding genes were validated by RT-qPCR. *PPP6R1*, *CCDC97*, and then either *ACTB* or *EPHA2* demonstrated the most stable mRNA levels. Normalizing target gene C_q_ data with at least three of these RGs simultaneously, as per MIQE guidelines (*PPP6R1* and *CCDC97* with either *ACTB* or *EPHA2*), resulted in congruent conclusions. *FABP5* expression was increased in ADs (5.99 and 8.00 fold, *p* = 0.00002 and *p* = 0.0003) and in OBs (5.18 and 5.91 fold, *p* = 0.0011 and *p* = 0.0023) relative to ADSCs. *RUNX2* expression was slightly higher in ADs relative to ADSCs (1.97 and 2.65 fold, *p* = 0.04 and *p* = 0.01), but not in OBs (0.9 and 1.03 fold, *p* = 0.58 and *p* = 0.91).

## 1. Introduction

To meet MIQE guidelines, at least three stable reference genes (RGs) are required for conducting comparative quantitative analysis of target gene (TG) expression levels in response to different conditions by reverse transcription quantitative real-time-polymerase chain reaction (RT-qPCR) [1]. The RGs are used to normalize the TG quantification cycle (Cq) values as part of the analyses to determine the fold change in TG expression under treated versus untreated conditions. Good RGs for RT-qPCR must demonstrate stable expression across all experimental conditions of specific interest. Multiple algorithms are available that can be used to assess stability.

Common RGs used for gene expression studies in the equine mesenchymal stem cell model are glyceraldehyde-3- phosphate dehydrogenase (GAPDH), β2-microglobulin (β2M), and β-actin (ACTB) [2,3,4,5]; however, it has been reported that these genes are not stably expressed in equine mesenchymal stem cells derived from adipose tissue as compared to those isolated from bone marrow [6]. Furthermore, it was reported as part of an analysis of over 225 RT-qPCR articles, published in prominent livestock journals from 2013–2017, that less than 10.7% of these articles used reliable RGs in the RT-qPCR experiments [7]. RT-qPCR assays are impacted by several variables that can affect the reliability of quantification, such as quality of the extracted RNA, primer design, and the efficiency of cDNA synthesis, as discussed in the MIQE guidelines and by Taylor et. al., 2019 [1,8]; however, an acceptable approach to minimizing the effect of such variations is to normalize the expression level of the TG to the expression of a stably expressed gene, called the RG, which is impacted by all variables that affect the TG, but is not significantly regulated in response to the test/treated condition [9,10].

Therefore it is necessary to identify reliable RGs for RT-qPCR specifically for the experimental condition of interest, and for this research, an equine adipose-derived mesenchymal stem cell (EADMSC) differentiation model. Several algorithms, including BestKeeper [9], NormFinder [11], geNorm [10], ΔCT [12], and RefFinder [13], can be used to assess the stability of candidate RGs. The BestKeeper algorithm determines the optimal RG through pair-wise correlation analysis of all pairs of candidate genes to identify genes with low variable expression, and genes with standard deviation values greater than one are considered unsuitable as RGs [9]. The NormFinder algorithm for RG selection is based on an estimation of the overall variation of the candidate RGs and the variation between sample subgroups of the sample set [11]. It directly measures the estimated expression variation so that the systematic error introduced when using the gene can be evaluated, and it calculates a stability factor for each candidate RG. The software then ranks the candidate genes based on the stability factor. In the case of the geNorm algorithm, the pair-wise variation for each candidate RG is determined with all other control genes. It is expressed as the standard deviation of the logarithmically transformed expression ratios to define a stability measure referred to as the M value, the average pairwise variation between a single candidate RG and all other control genes [10]. Candidate RGs with the lowest M values have the most stable expression in the tested samples, and candidates with an M value lower than 1 are considered suitable [10]. The ΔCT comparative method compares the relative expression of pairs of genes within each sample. If the ΔCT value between the two genes remains constant when analyzed in the different samples, it indicates that both genes are stably expressed in the samples, or both genes are co-regulated [12]. More RGs can be added into the comparison to provide more information on which pairs show less variability and which RGs are most stable in the samples tested. Finally, the web-based ReFinder algorithm integrates geNorm, Normfinder, BestKeeper, and the comparative ΔCT method to compare and rank the tested candidate RGs. It assigns an appropriate weight to an individual gene and calculates the geometric mean of their weights for the overall final ranking [13]. For this study all of the above mentioned algorithms were utilized to test the stability of candidate RGs.

Adipose-derived stem cells (ADSCs) in humans and equines are multipotent progenitors for many specialized cell types, including ADs, ODs, chondrocytes, and other cell types, and are used in stem cell therapies to support healing [2,14]. Therefore, we are interested in using the EADMSC differentiation model as an animal model for cell development, specifically to explore the impact of various external stimuli on gene expression during development. As a first step toward that goal, in this study, we set out to identify at least three RGs for which mRNA and protein expression levels remain constant during the process of equine ADSC differentiation into ADs or ODs, in order to meet MIQE guidelines, and as recommended by Vandesomple et. al., 2002 [1,10]. Based on this study, four RGs are recommended as suitable for the EADMSC model described here.

## 2. Materials and Methods

### 2.1. Cell Preparation and Treatments 

Adipose tissue samples were taken from a young adult (2–5-year-old) male horse cadaver, as described briefly here and in detail in [2]. Euthanasia was performed by sedation with xylazine hydrochloride (1.1 mg/kg, IV), followed by injection of phenobarbital sodium (108 mg/kg, IV). Adipose tissue was collected in vials of cold α MEM and either used immediately or stored on ice in a refrigerator at 4 °C for 12 h. Sterile chilled PBS was added to petri dishes to keep the tissue samples wet. The tissues were cut into 1 cm segments in the petri dishes, then added to 2 mL cryovials filled with 7.5% DMSO (dimethyl sulfoxide) in phosphate buffered saline (PBS). The vials were kept at room temperature for 30 min to allow the freezing medium to saturate the tissue. Next, the cryovials were inserted into polystyrene foam containers and kept in a −80 °C freezer for at least 24 h. After processing, all samples were stored in a liquid nitrogen tank (approximately −196 °C) within 72 h. The horse was donated to the Atlantic Veterinary College for another research project [2] and was euthanized with approval from the University of Prince Edward Island Animal Care Committee.

For cell culture preparations, adipose tissue samples were defrosted, and tissues were removed from vials using mosquito hemostats. The tissue was placed into 50 mL centrifuge tubes containing 25 mL of sterile PBS. For cell culture harvesting, the samples were gently mixed, and then the PBS buffer was removed by suction; the tissue sample was then washed once more with PBS. The tissue was then cut into smaller pieces. Ten milliliters of Collagenase Type I, 2000 units/mL (Invitrogen), was added. The tube was vortexed and incubated at 37 °C in 5% CO_2_ for 60 min in a Sanyo MCO-20AIC CO_2_ incubator, with intermittent mixing every 20 min. Next, ten milliliters of standard medium (SM, α-minimal essential medium (α-MEM), 10% fetal bovine serum (FBS; Gibco, 16140), 10,000 U Penicillin, 10 mg/mL Streptomycin, 2 mM L-glutamine, and 250 µg/mL Amphotericin B) was added. The digest was passed through a tea strainer to remove the remaining tissue pieces and then filtered with a 70 µm sterile cell strainer (Fisher Scientific, 352350). The cell strainer was then rinsed with standard medium (SM, α-minimal essential medium (α-MEM), 10,000 U Penicillin, 10 mg/mL Streptomycin, 2 mM L-glutamine, and 250 µg/mL Amphotericin B) with 10% FBS. The filtered digest was spun at 1500 RPM for 10 min, and the supernatant was removed. The number of viable cells harvested was determined by 0.4% Trypan Blue exclusion and counted on a hemocytometer. These primary cells were then maintained in a sterile flask at 37 °C and 5% CO_2_ in SM with 10% FBS.

### 2.2. Cell Culture and Experimental Design

The primary cells that had attached to the surface of the sterile flasks were detached from the flask surface with 10 mL of a 1:4 dilution of 0.25% Trypsin-EDTA (Gibco, 25200-072) into a cell stripper dissociation reagent (Corning™ 25056CI) and then incubated for 30 min at 37 °C. The cells were expanded to two passages and then used for proteomic and gene expression studies, as described:

Three replicates for the adipogenic medium (DMEM/F12, Gibco, 11320, Thermo Fisher Scientific, Mississauga, Ontario, Canada, 5% FBS (Gibco, 16140, Thermo Fisher Scientific, Mississauga, ON, Canada), 3% Rabbit serum (Invitrogen, Toronto, ON, Canada, 10510), 10,000 U Penicillin, 10 mg/mL Streptomycin, 2 mM L-glutamine, 250 µg/mL Amphotericin B, 33 μM Biotin (Sigma, Oakville, ON, Canada, B4639), 1 µM Pantothenate (Sigma, Oakville, ON, Canada, P5155), 20 nM Dexamethasone, 0.5 mM IMBX (3-isobutylmethylxanthine) (Sigma, Oakville, ON, Canada, I7018) and 5 µM Rosiglitazone (Toronto Research Chemicals, Toronto, ON, Canada, R693505), prepared as per [2]) condition and three for the SM condition (untreated) were plated into one six-well plate at a density of 1.15 × 105 cells/well, and three replicates for the osteogenic medium (OM, SM recipe plus 50 µg/mL Ascorbic acid, 10 mM β-glycerophosphate, 10^−8^ M Dexamethasone as per [2]) condition. In addition, three replicates for the SM condition were plated in a second six-well plate. Both six-well plates were incubated in 5% CO_2_ at 37 °C for ten days.

The first six-well plate, containing the ADM condition, was stained with oil red O stain to detect lipid droplets. Cells grown in SM or ADM were fixed in 10% neutral buffered formalin and then incubated for 15 min at room temperature. The solution was removed with suction, then rinsed with distilled water, and left for 15 min. Next, two milliliters of 60% isopropanol were added, and the cells were incubated for 5 min at room temperature. The solution was removed with suction, and the cells were dried at room temperature. Next, one ml of Oil Red O working solution (0.35% in isopropanol) was added to the dry cells and incubated at room temperature for 10 min. The solution was removed with suction and rinsed with distilled water. The cells were then washed four additional times with distilled water.

The second six-well plate, containing the OM condition, was stained with Von Kossa stain to detect calcium deposits. Cells grown in SM or OM were fixed in 10% neutral buffered formalin and then incubated for 15 min at room temperature. The solution was removed with suction, then rinsed with distilled water, and left for 15 min. The cells were then stained with 2.5% silver nitrate for 30 min during the Von Kossa staining procedure. Next, the solution was removed with suction and then rinsed with distilled water. Next, sodium carbonate formaldehyde was added for 5 min, and the solution was then removed with suction and rinsed with distilled water.

Pictures of the stained cells were acquired with a Zeiss Axiovert 40 CFL Trinocular Inverted Fluorescence Phase Contrast Microscope with 20× objective lens. Photo brightness and contrast were adjusted with Adobe Photoshop 2020.

Two additional six-well plates (12 wells) were seeded with ADSCs for each of the three conditions SM, ADM, and OM. Cells harvested from two wells for each condition were pooled, resulting in six biological replicates for each condition. Three biological replicates for each experimental condition were stored at −80 °C for DNA extraction, and RNAlater was added to the remaining three biological replicates for each of the three conditions for RNA extraction. After overnight saturation at 4 °C, the three biological replicates, for RNA extraction were stored at −80 °C until further processing.

Eighteen additional ADSC cell culture replicates were prepared in T75 flasks at a density of 1.2 × 10^4^ cells/cm^2^. Of these, six replicates were incubated in SM, six in ADM, and six in OM. Cells harvested from two T75 flasks for each condition were pooled for protein extraction, resulting in three biological replicates for each condition. The proteins extracted were subjected to proteomic analysis by liquid chromatography tandem mass spectrometry (LCMSMS and Proteome Discoverer 2.2 were used to identify trypsin-digested peptides detected based on the Uniprot Database Equine caballos protein repository [9796]).

### 2.3. Cell Harvesting and Storage

The cells were harvested for protein and mRNA extraction by washing each flask 3 times with 10 mL PBS, adding 5 mL PBS, scraping the cells from the flask surface, and then adding the cells to a 10 mL tube that was centrifuged at 2500× *g* for 10 min. Then, the supernatant was removed, and the cell pellets were each added to a 1.5 mL Eppendorf tube for further processing.

For cells used for mRNA extraction, RNAlater™ Stabilization Solution (AM7022) was added, and cells were stored as previously described. Cells used for protein extraction were directly stored at −80 °C.

### 2.4. Sample Preparation for LCMS3

For the LCMS3 analysis, proteins were prepared by methanol-chloroform precipitation and standard chromatography methods for protein separation, as described. The cells were washed three times with 20 mL sterile PBS each time. The cells were then suspended in 5 mL of sterile PBS, uniformly scraped, and transferred to 15 mL centrifuge tubes. This process was repeated three times, and then the flask was washed with 5 mL of PBS and added to the tubes. The cells were centrifuged at 2500× *g* for 10 min, and the supernatant was siphoned off. Up to 200 µL Lysis Buffer (10 mM Tris HCl at pH 8.0, 100 mM NaCl, 1 mM EDTA, 2% SDS was added plus 7 µL Protease inhibitor (Sigma-Aldrich, St. Louis, MO, USA, P8340) per 7 × 10^6^ cells and 1 mM DTT). A Fisher Sonic Dismembrator, model 300, was used for sonication, in 3 cycles of 20 s on and 10 s off, on ice, to lyse the cells. The probe tip was washed with 70% isopropanol and dried with a wipe between samples. The lysate was then clarified by centrifugation, 5000× *g* for 20 min at 4 °C. The proteins were then precipitated from the cell lysate, using the following chloroform/methanol precipitation protocol.

To a sample with a starting volume of 200 µL, 800 µL of methanol, 200 µL of chloroform, and 600 µL of mass spectrometry-grade water were added, mixing thoroughly after each step. The sample was then centrifuged for 1 min at 14,000× *g*. The top aqueous layer was then removed, and 800 µL of methanol was added and mixed thoroughly. The samples were then centrifuged for 2 min at 14,000× *g*. Finally, the methanol layer was removed, and the pellet was allowed to dry. The resulting pellet was suspended in 8 M urea buffer for LC-MS3: 0.4 M NH_4_ HCO_3_: 0.1% SDS and stored at −80 °C.

Protein concentrations of the samples were determined (Pierce™ BCA protein assay kit, 23227, Thermo Fisher Scientific, Mississauga, ON, Canada). An additional pooled sample was prepared by adding one microliter from each sample into a separate tube. Samples and pools were normalized in the urea buffer to yield a final volume of 200 µL and 100 µg of total protein. Samples were then reduced with 10 µL 0.5 M DTT (Fisher BioReagents: BP172-5, Fisher Scientific, Hampton, NH, USA) and incubated at 60 °C for 30 min. Next, 20 µL of 0.7 M iodoacetamide (Sigma, Oakville, ON, Canada, I1149) was added, and the samples were incubated for 30 min at room temperature. The samples were then diluted with 1.2 mL of Millipore water, then digested with 100 µL of 0.2 µg/µL buffered Trypsin (20 µg Pierce™ Trypsin Protease (Thermo Fisher Scientific, Mississauga, ON, Canada, 90657) in 50 µL trypsin dilution buffer (Promega, Madison, WI, USA, V542A) and incubated at 37 °C overnight. The following morning, an equivalent amount of Trypsin was added and incubated for two hours at 37 °C. 

The pH for the protein samples was decreased to below three by adding Formic acid (Fisher: A117-50) and TFA (Trifluoroacetic Acid; Fisher Scientific, Hampton, NH, USA, A117-50). Liners (Supelco, Sigma, Oakville, ON, Canada, PN 57059) and columns were then added to a Supelco VisipropDL manifold. The columns (Supelco, Sigma, Oakville, ON, Canada PN WAT094226) were washed once with 0.5 mL of 100% methanol (Fisher Chemical, Fisher Scientific, Hampton, NH, USA A456-4), once with 1 mL of 50% acetonitrile (Honeywell, 34967) + 0.1% TFA, and once with 1 mL of 0.1% TFA. The protein samples were then loaded onto the columns and washed four times with 1 mL of 0.1% TFA. The proteins were then eluted from the column by washing twice with 0.5 mL of 50% acetonitrile + 0.1% TFA and dried using an SPD Speed-Vac (Thermo, SPD11V).

Sample pellets were then suspended in 70 µL 100 mM HEPES buffer (Gibco, Thermos Fisher, Mississauga, ON, Canada, 11344041). The TMT10plex Isobaric Reagent Set (Thermo Scientific, 90110) was suspended per the manufacturer’s instructions. Thirty microliters of each reagent was added to the samples and to the pool as shown in Table 1; the reaction was incubated at room temperature for one hour.

The reactions were each quenched with 5 µL 5% hydroxylamine (Sigma-Aldrich, 159417). Samples were then pooled, and the combined samples were then diluted 1:15 times with 0.1% TFA. Finally, the solid phase extraction process was repeated to desalt the combined sample. 

The combined samples were suspended in Buffer A (95% water, 5% Acetonitrile, 10 mM NH_4_HCO_3_, pH 9.0). These samples were then separated by reverse phase liquid chromatography using an ÄKTA pure (GE Healthcare Life Sciences, Chicago, IL, USA) at high pH using a 100 × 4.6 mm Onyx Monolithic C18 column (Phenomenex, Torrance, CA, USA, CH0-7643), with a flow rate of 1 mL/minute. The column was run for 15 min at a gradient of 0–40% of Buffer B (5% water, 95% Acetonitrile); for Buffer A, the column was run for 5 min up to 100% Buffer B. Sixty fractions of 0.6 mL volume were collected, then the fractions were concatenated by combining fractions 1–16 and 31–46; 2–17 and 32–47, etc., into 15 samples labelled 1, 2, …, 15.

The samples were transferred to 300 µL HPLC vials and subject to analysis by LC-MS/MS on a Thermo Scientific Orbitrap VelosPRO mass spectrometer equipped with an UltiMate 3000 Nano-LC system (ThermoFisher Scientific, Mississauga, ON, Canada). Chromatographic separation of the digests was performed on a PicoFRIT C18 self-packed 75 mm × 60 cm capillary column (New Objective, Woburn, MA, USA) at a flow rate of 300 nL/min. MS and MS/MS data were acquired using a data-dependent acquisition method in which a full scan was obtained at a resolution of 30,000, followed by ten consecutive MS/MS spectra in collision-induced dissociation (CID) mode. The most intense peak from MS/MS was subjected to MS3 by higher-energy collisional dissociation (HCD) (normalized collision energy of 36%) to scan the low-mass TMT reporter ion region. Internal calibration was performed using the ion signal of polysiloxane at *m*/*z* = 445.120025 as a lock mass.

Raw MS data were analyzed using Proteome Discoverer 2.2 (ThermoFisher Scientific, Mississauga, ON, Canada). Peak lists were searched against the UniProt Equus Caballus Knowledge Base-database and the cRAP database of common contaminants (Global Proteome Machine Organization). Cysteine carbamidomethylation was set as a fixed modification, while methionine (Met) oxidation, N-terminal Met loss, and phosphorylation on serine, threonine, and tyrosine were included as variable modifications. A mass accuracy tolerance of 5 ppm was used for precursor ions, while 0.02 Da for HCD fragmentation, or 0.6 Da for CID fragmentation was used for product ions. A percolator was used to determine confident peptide identifications using a 0.1% false discovery rate (FDR). Site-specific determination of phosphorylated amino acids was confirmed using PhosphoRS (ThermoFisher Scientific, Mississauga, ON, Canada). TMT quantifications were performed on the TMT reporter ions from the MS3 scans.

### 2.5. Selection of Candidate RGs from Proteome Data

Relative abundance units of proteins were obtained by LCMS3 for the three replicates for each sample type, progenitor ADSCs, OBs, and ABs. Analysis of variance (ANOVA) was conducted on these relative protein abundance values, for each protein detected. In previous experiments with other potential candidate RGs, ADs demonstrated the greatest variance in gene expression as compared to OBS and progenitor ADSCs (data not shown); therefore, the means for protein abundance for proteins extracted from ADs were subtracted from the means for protein abundance for proteins extracted from progenitor ADSCs, to identify proteins that were most stably expressed across the adipocyte and progenitor ADSC samples. Proteins were ranked by ascending order, based on this difference in the means so that those with the smallest difference in means represented the most promising candidates. 

The first 25 accession numbers in the ranked list were cross-referenced to ANOVA *p*-values to confirm a *p*-value > 0.5, indicative of no significant difference between the means, and 23 proteins were identified that had a *p*-value > 0.05. The difference of means between the progenitor ADSCs and OB samples, and the difference of means between the AD and OB samples were also calculated for these 23 proteins to ensure that there was little variance between all paired means. Uniprot accession numbers were cross-referenced to NCBI gene accession numbers, if records were available (Table 2).

### 2.6. Primer Design 

As indicated in Table 3, gene records were imported into Generunner software, and primers pairs were selected for high stringency based on the following parameters: TM ≥ 56 °C, product size ≥ 70 base pairs (bps) and ≤ 200 bps, and primer size between 18–22 bps. Primers with a combination of high annealing temperatures and low differential temperatures between primers were priority selected. Primers were designed to flank the intron splice sites (Table 3). Designing primers that flank the intron splice sites allowed us to control for gDNA contamination, since gDNA contamination of RNA samples would result in the amplification of at least two different sized products. The selected primers were individually used in blast analysis to confirm the specificity of annealing within the Equus caballus gene records at NCBI. Primers that showed high specificity for single sites in the equine genome were selected. Primer specificity was confirmed by PCR amplification, using cDNA generated from RNA that was isolated from the progenitor EADMSCs as template, followed by Sanger sequencing of the amplicons generated.

### 2.7. Determination of Optimal Annealing Temperatures for Selected Primers

In order to select the optimal annealing temperature for each primer set, PCR reactions were run using a temperature gradient ranging from 60 °C to 75 °C, with even temperature increments spanned across twelve thermocycler wells to identify the temperature at which a single specific product was produced with the highest yield, using a Biometra T-gradient thermoblock. PCR products were visualized by agarose gel electrophoresis. Primers were used to amplify their respective gene targets at their optimal annealing temperature using the following program: 1 cycle: 2 min at 94 °C; 35 cycles: 1 min at 94 °C, 30 s at the appropriate annealing temperature, 30 s at 72 °C; 1 cycle: 5 min at 72 °C using a BIORAD C1000 Touch thermal cycler. The PCR products were sequenced at the Center for Applied Genomics (Toronto Sick Kids Hospital), and sequences obtained were subjected to nucleotide Basic Local Alignment Search Tool (blastn) analysis (National Center for Biotechnology Information (NCBI)) to confirm PCR product identity and primer specificity.

### 2.8. Isolation of RNA and Synthesis of Complementary DNA

A standard protocol was followed according to the AurumTM Total RNA Mini kit (Bio-Rad, Hercules, CA, USA) for RNA extraction from cell culture samples. Briefly, cells attached to six-well plates were washed with PBS three times, then transferred to a micro-centrifuge tube, and lysed with a lysis solution. Ethanol was added to allow binding in the column. After binding, samples were washed and treated with RNAase-free DNase 1, for 15 min at room temperature. Samples were then washed twice with ethanol prior to elution in buffer. cDNA was synthesized using a standard 100 ng of RNA into a total reaction volume of 20 μL for each biological replicate. RNA was quantified using a NanoDrop 1000 spectrophotometer, and RNA quality was assessed based on the A260/A280 ratio, which approximated a value of 2 for the individual samples. RNA samples from each biological replicate were converted to cDNA independently, following qScript cDNA SuperMix protocol, and incubated under the following conditions: 5 min at 25 °C, 30 min at 42 °C, and 5 min at 85 °C. Multiple cDNA reactions were generated for each biological replicate, and these multiple reactions were pooled for each biological replicate in order to generate sufficient cDNA for the generation of standard curves. The cDNA concentrations obtained in all cases was above the highest stock concentration selected for the standard curve (960 ng/µL); therefore, the cDNA pools generated for each biological replicate were standardized to 960 ng/µL.

### 2.9. Quantitative Real-Time PCR

The iQTM SYBR^®^ Green Supermix (Biorad, CA, USA) was used for all qPCR reactions. Briefly, each reaction consisted of 7.5 µL template, 1.25 µL each for forward and reverse primers (10 µM), and 10 µL IQ™SYBR^®^ Green supermix (2×) in a final reaction volume of 20 µL. For generation of standard curves for each primer set, 2-fold dilution series were performed to generate a series of cDNA concentrations ranging from 960 ng/µL to 15 ng/µL. A standard volume of 7.5 µL of each of these concentrations was added to individual reactions, with a final volume of 20 µL, resulting in 7 different concentrations, ranging from 360–5.625 ng cDNA/µL in the final reactions. Standard curves were used to determine the template concentrations at which primer efficiency for the candidate RGs and TGs would be most efficient. Two sample points at either end of the range were dropped as necessary to identify the concentration range at which primer efficiency was optimal. A minimum of 5 sample points were used to generate each efficiency curve. Primer efficiency was calculated from the standard curve slope generated, using a LightCycler 480 instrument version 1.5 software. The data was analyzed using the second derivative max method to determine Cq values, and the melting curves for the qPCR products generated were assessed using TM Calling analysis to confirm primer specificity (LightCycler 480 instrument version 1.5 software). Melting curves were run to confirm the presence of a single amplicon as a control for gDNA contamination or other circuitous contaminating templates.

PCR efficiency (E) was calculated with the formula E = 10(−1/slope). Percentage efficiency was calculated with the formula (E−1) × 100%. The coefficient of determination (R^2^) was reported (Table 3).

### 2.10. Analysis of TG Expression 

The mean Cq values for three technical replicates for each of three biological replicates for ADSCs, ADs, and OBs for the different reference and TGs was obtained from the RT-qPCR report generated using Lightcycler 480 instrument version 1.5 software. This data was entered into a spreadsheet (See Appendix A). The Pfaffl method [15] and the methods described by Vandesompele et al. (2002) [10] and Hellemens et al. (2007) [16] were used to determine the relative gene expression, since up to three RGs were used for TG data normalization, accounting for the differences in primer efficiencies. The method described by Vandesompele et al., (2002) and Hellemens et al., (2007) is very similar to the Pfaffl method, with the key difference being that the former method allows for normalization using more than one RG, by geometric averaging of all the relative quantities (RQ) of the multiple RGs, as described in detail in references [17,18]. Two-tailed T-tests were performed for each TG to compare the difference in gene expression between the control (ADSCs) sample and each of the treated (ADs and OBs) samples (Appendix A).

The Pfaffl method [15] was used to calculate each TG’s relative gene expression based on normalization with individual RGs, using the samples’ relative quantity (RQ) values, calculated as described. A two-tailed T-test was performed to compare the difference in relative gene expression in ADSCs (untreated conditions) as compared to ADs (treated condition 1) or OBs (treated condition 2) for each TG, as determined by normalization with each RG (Appendix A).

### 2.11. Determination of Protein Fold Differences

The relative protein abundances for each selected protein for the three experimental conditions (SM, ADM, and OM) were obtained by LCMS3. First, the relative abundances for each protein for three biological replicates for the untreated control condition (SM) were averaged. Then, the individual protein abundance ratio for each biological replicate divided by the average abundance for the untreated condition was calculated to determine the individual fold change for each biological replicate. These ratios were averaged for each experimental condition SM, ADM, and OM and are reported as fold change. Finally, a two-tailed T-test was performed to assess the significance of the fold-change in protein expression measured at the *p* = 0.05 level (Appendix A).

## 3. Results

### 3.1. Validation of ADSCs Differentiation into ADs and OBs

Briefly, three replicates for the ADM condition and three for the SM condition (untreated) were plated into one six-well plate. Three replicates for the OM condition and three for the SM condition were plated in a second six-well plate. Both six-well plates were incubated in 5% CO_2_ at 37 °C for ten days. For the ADM condition, the first six-well plate was stained with oil red O stain to detect lipid droplets, and the second six-well plate, for the OM condition, was stained with Von Kossa stain to detect calcium deposits. As expected, cells cultured in ADM were stained red, indicating the presence of oil droplets, and cells cultured in OM had accumulated a dark foci due to the staining of calcium deposits with silver nitrate, as shown in Figure 1, thereby validating the ADSC cell differentiation process.

Proteins extracted as described from ADSCs, ADs and OBs were subjected to proteomic analysis by LCMS3, and Proteome Discoverer 2.2 was used to identify trypsin-digested peptides detected based on a search of the Uniprot Database *Equuscaballos* protein repository [9796]. The nine cell cultures (three biological replicates each for the ADM, OM, and SM conditions) were all positive for the mesenchymal stem cell markers, Runx2 and Cd44, as determined by LCMS3 (Figure 2). Furthermore, Fabp5 protein levels were only significantly expressed at higher levels in ADs as compared to ADSCs, and alkaline phosphatase (Alp) protein levels were only significantly expressed at higher levels in OBs, as compared to ADSCs, as was expected for the different cell types (Figure 2), further validating ADSC differentiation into ADs and OBs.

### 3.2. Candidate RG Selection for RT-qPCR

For this study, seven genes (*PPP6R1*, *EPHA2*, *CCDC97*, *EHD3*, *ACTB*, *β2M*, and *GAPDH*) were selected and evaluated as potential RG candidates, and primers for their amplification were designed. Primer sequences and amplicon sizes are reported in Table 3. The PCR products were evaluated by gel electrophoresis for amplicon size, DNA sequencing for identity confirmation, and RT-qPCR melting curves for specificity (data not shown). Primer efficiencies ranged from 83.65% to 109.42%, as determined by RT-qPCR (Table 3). Three of these genes are the standard RGs used in equine studies, including *ACTB*, *β2M*, and *GAPDH* [2,3,4,5]. The other four genes were selected based on an analysis of proteomic data, as described. First, the analysis of variance (ANOVA) among means of the relative abundances of all proteins identified among the three conditions (3 biological replicates for each condition) was analyzed. Next, proteins that did not demonstrate a significant difference in the means among the three conditions (SM, ADM, and OM) (*p* > 0.05) were filtered. Subsequently, the differences in means between each condition (AD-SM, AD-OB, and OB-SM) for these proteins were calculated and ranked from smallest to largest difference, based on the AD-SM comparison. The top 23 proteins, their *p* values, and the differences in the mean relative abundances for these proteins between the three cell types are reported in Table 2. A selection of these proteins, for which gene sequence data were available, and for which the difference in means between the AD-OD comparison was also low, were selected for validation by RT-qPCR, including (EHD3, TRIM26, LRSAM1, CCDC97, PPP6R1, and EPHA2); however, primers that specifically amplified a single, specific target and that flanked mRNA splice junctions could only be designed successfully for *EHD3*, *CCDC97*, *PPP6R1*, and *EPHA2* (Table 3). Relative protein abundances across the SM, ADM, and OM conditions for these candidate RGs are reported in Figure 3 (See Appendix A for calculations). None of the four proteins demonstrated any significant difference in protein expression in the ADM or OM conditions relative to the SM condition (*p* > 0.05). The β-actin relative protein abundance levels are also reported in Figure 3.

### 3.3. RG Stability Profiles

RG candidates’ stabilities were assessed across the three experimental conditions, SM, ADM, and OM, using five different algorithms, including Bestkeeper, NormFinder, geNorm, ∆ C_T_ analysis, and RefFinder analysis (Table 4). Based on Bestkeeper, which sets a cut-off of acceptability at a standard deviation of 1 among the C_q_ values for all nine samples tested for each RG candidate, all RGs tested met the suitability criteria; however, based on the pairwise Pearson correlation coefficient (r) between primers, *CCDC97*, *PPP6R1*, and *GAPDH* were ranked as the most stable candidates, since they had the highest r values (0.943, 0.883 and 0.769, respectively). Among the other algorithms, including NormFinder, geNorm, ∆ C_T_^,^, and RefFinder analysis, *PPP6R1* and *CCDC97* were consistently ranked as the most stable RGs. However, there were some inconsistencies among the algorithms regarding which RG was considered the next-most stable. Since RefFinder analysis integrates geNorm, Normfinder, BestKeeper, and the comparative ΔC_T_ analyses, assigning appropriate weight to an individual gene and calculating the geometric mean of their weights for the overall final ranking, RefFinder’s rankings were used to select the third-most stable RG for the equine EADMAC differentiation model, *ACTB*. *EPHA2* was selected as an alternate RG for *ACTB*, since after *PPP6R1* and *CCDC97*, it was the next most consistently ranked as the third- or fourth-most stable across the experimental conditions tested by the five algorithms and was also ranked as the third-most stable RG as per geNorm.

Furthermore, geNorm analysis demonstrated an average M expression stability of less than 1 for all candidate RGs across all samples, also indicating, similar to the Bestkeeper algorithm, that all tested RG candidates are stable enough to be used as RGs. geNorm additionally calculates a pairwise variation value (V) to determine the optimum number of RGs required for accurate normalization of TG Cq values for a given experimental situation. Normalization factors were calculated for the three most stable control genes (those with the lowest M value, where M is defined as the average pairwise variation of one gene with all the other control genes) and for four additional internal control genes by stepwise inclusion in order from the most stable to least stable. Pairwise variation (V) was subsequently calculated for every series of NFn and NFn + 1 to determine the optimal number of RGs required for normalization. Based on this analysis (Appendix A), the V value dropped below the recommended 0.15 cut-off when comparing a normalization factor based on the second- or third-most stable RGs (V2/3 < 0.15), indicating that for this experimental situation, two RGs are sufficient for normalization of TG RT-qPCR data. However, to meet MIQE guidelines and Vandesompele et al. (2002) recommendations [1,10], three RGs were used to normalize TG RT-qPCR data.

### 3.4. mRNA Expression profiles of TGs in ADSCs, Adipocytes, and Osteoblasts 

RG C_q_ values were used individually [15], or the geometric means of the RQ values for three genes were combined [10,16]—*PPP6R1*, *CCDC97*, and *ACTB*, or *PPP6R1*, *CCDC97*, and *EPHA2*—to normalize TG C_q_ values to determine fold-change differences between the SM and ADM condition or the SM and OM condition by the comparative C_T_ method (Figure 3, and Appendix A). *FABP5* mRNA expression levels were significantly higher in the ADM versus the SM condition and in the OM versus the SM condition when normalized with individual RGs or RG combinations (Figure 4). *RUNX2* mRNA expression levels were also elevated in the ADM versus SM condition when normalized with individual and RG combinations, except for *CCDC97* (*p* = 0.397); mRNA levels were not significantly different between the OM and SM conditions (Figure 4).

## 4. Discussion

### 4.1. RG Congruence

Based on stability analysis, using Bestkeeper and geNorm, all RG candidates tested demonstrated sufficient stability to be used as RGs in an EADMSC differentiation model, as all genes demonstrated a standard deviation of less than 1 (BestKeeper), as well as M stability values of less than 1 (geNorm). In almost all cases, when either individual RGs or the combined geometric means of the RQ values for at least three RGs were used to standardize RQ values, the expression data were congruous for RUNX2 and FABP5 mRNA expression. However, in one case, when CCDC97 alone was used to normalize RUNX2 Cq data from ADs relative to ADSCs, the analysis indicated no significant difference in mRNA expression values. In contrast, all other normalization methods indicated a significant difference. However, when CCDC97 RQ data was used in combination with either PPP6R1 and ACTB or PPP6R1 and EPHA2 RQ data, the fold-change detected for RUNX2 between ADs and ADSCs was consistently found to be significant. This observation highlights the importance of using multiple RGs, at least three, to normalize TG expression as per MIQE guidelines [1] and using algorithms such as geNorm to determine the optimal number of RGs necessary for each unique experimental situation.

### 4.2. TG Expression

Fabp5 is a fatty acid-binding protein that is expressed primarily in adipocytes and macrophages [19,20]. Furthermore, Fabp5 is reported to be a transcriptional and metabolic regulatory factor that is secreted by adipocytes [19]. In this study similar trends were observed in that FABP5 mRNA levels were elevated significantly in differentiated adipocytes as compared to the progenitor ADSCs, when normalized with either set of reference genes (5.99 and 8.00 fold, *p* = 0.00002 and *p* = 0.0003, Figure 4); furthermore, this was validated in that Fabp5 protein expression was found to be significantly increased in protein extracts taken from differentiated adipocytes as compared to ADSCs obtained by proteomic analysis (5.29 fold, *p* = 0.0026, Figure 2). FABP5 mRNA levels were also elevated in osteoblasts (5.18 and 5.91 fold, *p* = 0.0011 and *p* = 0.0023, Figure 4), although protein levels ultimately remained at the same level in OBs compared to ADSCs (Figure 2). Perhaps in the case of OBs, Fabp5 expression was downregulated at the protein level as opposed to at the transcriptional level.

Runx2 protein expression levels were higher in ADs and OBs than ADSCs (1.91 fold and 1.90 fold, *p* = 0.0001 and *p*= 0.0013, respectively, Figure 2). RUNX2 mRNA levels were also increased in ADs compared to ADSCs, when normalized with either set of reference genes (1.97 and 2.65 fold, *p* = 0.04 and *p* = 0.01), but not OBs compared to ADSCs (Figure 4). It is reported in the literature that while Runx2 is expressed at higher levels in preosteoblasts, Runx2 expression wanes in mature osteoblasts [21]. In this model, OBs were harvested at only a single time-point that coincided with the positive staining of calcium deposits indicative of mature osteoblasts; hence, it is not surprising that at this sample point, during the later stage of osteoblasts development, that RUNX2 mRNA levels were not significantly higher in OBs as compared to ADSCs, and that protein levels were only slightly higher, less than two-fold higher (1.90 fold, *p* = 0.0013), in the OBs as compared to the ADSCs. Similarly, RUNX2 expression in ADs was less than two-fold higher (1.91 fold, *p* = 0.0001) and is not expected to be highly elevated in ADs compared to ADSCs, since it is reported that RUNX2 expression is reduced during adipocyte differentiation in response to dexamethasone treatment in a 72 h differentiation model [22].

### 4.3. The Impact of Primer Design and Experimental Model Specificity for RG Selection

In a paper by Nazari et al. 2015 [6], RGs GAPDH, ACTB, and β2M were not stable in equine adipose-derived stem cells compared to bone marrow-derived stem cells. Therefore, for this study, GAPDH, ACTB, and β2M primers were redesigned. Using these new primers (Table 3), the expression of these candidate RGs was shown to be stable across the experimental conditions tested, where ADSCs were propagated in the SM, ADM, and OM experimental conditions. Blast analysis using ACTB primer sequences from Nazari et al. 2015 [6] demonstrated that the ACTB forward and reverse primers are equally specific for both β and γ actin. They are designed to measure both actin variants (XM_023655002.1 and XM_023651796.1, respectively). In contrast, the ACTB forward primer designed for this study shares 100% identity with ACTB (XM_023655002.1) and only 81% identity with ACTG1 (XM_023651796.1), and the reverse primer is not specific for γ actin, so that this primer set detects only one actin variant. Differences in primer specificity may impact stability data results. As such, specificity for isoforms must be considered when designing primers, especially since isoforms such as β and γ actin are both ubiquitously expressed in many cell types, where they have different functions [23]. While differences in primer specificity could be a factor in differences in gene expression for ACTB measured in both studies, an alternative explanation for the difference in RG stabilities between this study and Nazari et al. 2015 [6] may be the differences in experimental conditions. In Nazari et al. 2015 [6], the experimental comparator for ADSCs was bone-derived stem cells, while this study’s experimental comparator was ADs or OBs. This finding highlights the importance of validating candidate RGs for stability for one’s specific experimental situation of interest and the risks of false results associated with selecting commonly used references for experimental studies without validating stability across the specific experimental conditions.

A limitation of this study is that mRNA and protein levels were only assessed at one time point, after incubation at 37 °C for ten days, once ADSCs had differentiated into mature ADs and mature OBs. Evaluating mRNA and protein expression levels at different time points during ADSC differentiation would have provided more information regarding the timeline for the stability of the RGs tested and provided more information on the dynamic profile for TG expression.

Another limitation of this study is that biological replicates were defined as cell culture replicates taken from a single horse, although derived from a heterogeneous primary ADSC population. In the future, the stability of the RGs identified will be tested in cells taken from other horses. However, this study has been useful in identifying candidate RGs for the EADMSC cell culture model that others may test in their unique experimental situations in studies that involve ADSC differentiation into ADs or OBs.

Another limitation of this study is that not all potential RG candidates identified by proteomics could be tested at the mRNA level for stability, since the equine genome is not fully annotated, and the intron splice sites for some genes are not known; thus, the primers could not be designed to flank the splice sites. As a result, we were only able to test some candidates identified by proteomics, and in some cases, for example for PPP61R, the best primers we could design had an R^2^ value 2% lower than optimal (88% versus 90%). However, PPP6R1 was consistently ranked as the most stable expressed RG across the conditions tested and is, therefore, recommended as a good choice for normalizing target gene data in the EADMSC model.

## 5. Conclusions

Proteomic evaluation proved an effective method to identify RG candidates for RT-qPCR in this EADMSC differentiation model. In this study, 100% (4 of 4) of the RG candidates identified by this method and then validated by RT-qPCR, including PPP6R1, EPHA2, CCDC97, and EHD3, were suitable for use as RGs in an EADMSC differentiation model. Furthermore, compared to other RGs most commonly used in this model (GAPDH, ACTB, and β-2M), PPP6R1 and CCDC97 were consistently ranked among the top most stable genes, followed by ACTB and EPHA2, although GAPDH, ACTB, and β-2M were still found to be sufficiently stable to be used as RGs in gene expression across the conditions tested. We recommend that a panel of stable RGs be validated by RT-qPCR for targeted gene expression studies before conducting comparative CT analysis [10,15,16].

### Furture Studies

We propose that the well-characterized EADMSC differentiation model described and the set of reference genes identified in this study are foundational tools that will be useful for studying the process of cell development and tissue repair in equines. In particular, we are interested in using this model to investigate the impact of stress on gene expression. For example, we have demonstrated that at the proteomic level, heat stress can have an impact on the intracellular morphological development of equine osteoblasts [24]. We would like to further explore the process of cell development under stressed conditions at both the mRNA and epigenomic levels.

## Figures and Tables

**Figure 1 genes-14-00673-f001:**
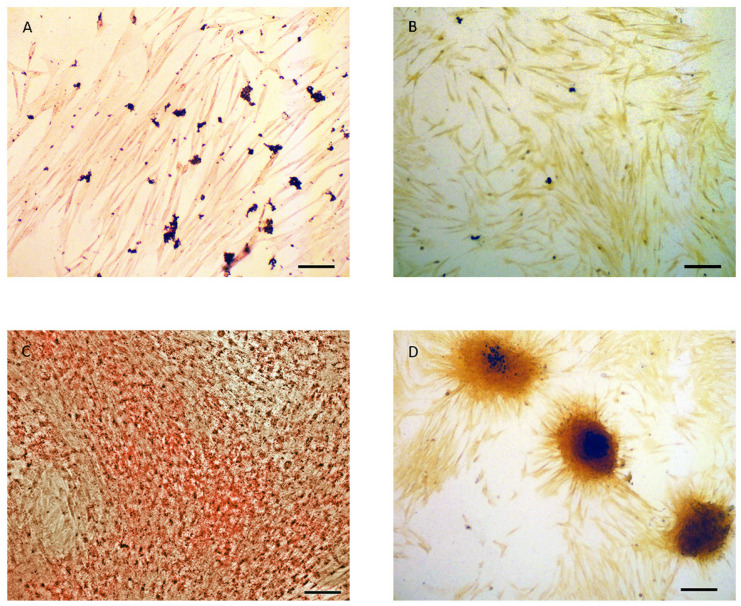
Oil Red O stained of ADSCs (**A**) and ADs (**C**), and Von Kossa stained ADSCs (**B**) and OBs (**D**). Size bars represent 135 µm.

**Figure 2 genes-14-00673-f002:**
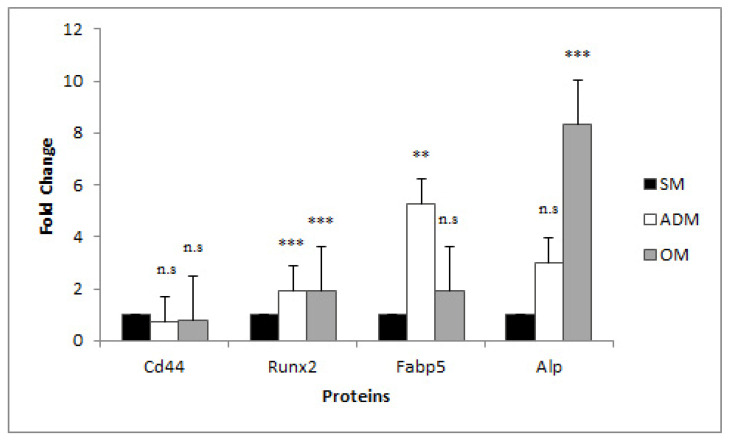
Relative expression of Cd44, Fabp5, Runx2, and Alp proteins in ADSCs submitted to SM, ADM, or OM conditions. ** = *p* < 0.01, *** = *p* < 0.001, n.s. = not significant.

**Figure 3 genes-14-00673-f003:**
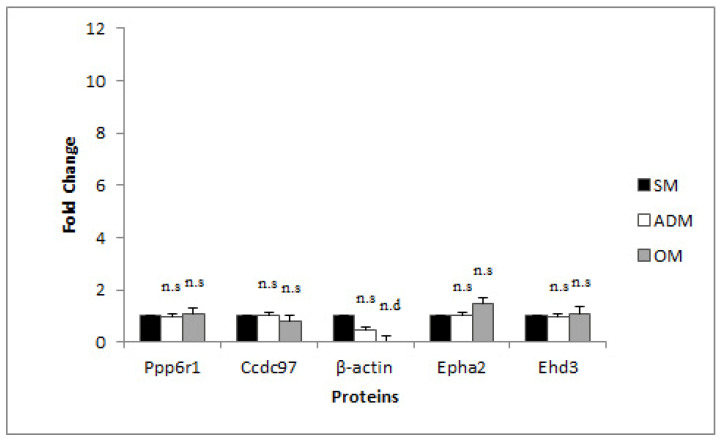
Relative protein expression for RG candidates selected by evaluation of ADSC, AD, and OB proteomes; the relative protein expression across conditions for β-actin is also reported. n.s. = not significant, n.d. = not detected.

**Figure 4 genes-14-00673-f004:**
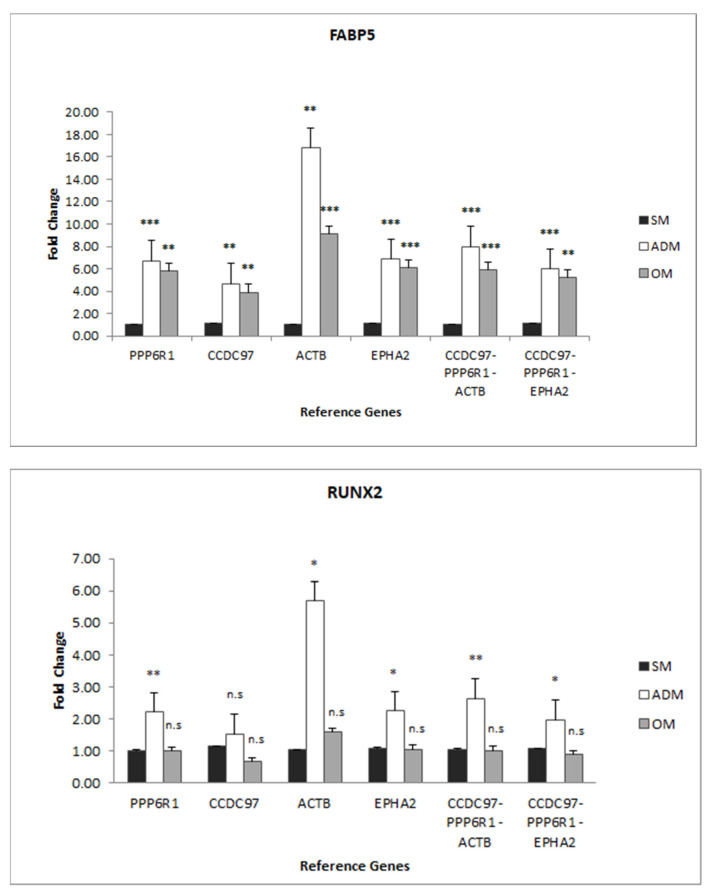
Relative expression of TGs, *FABP5*, and *RUNX2* in ADSC cells in response to SM, ADM, or OM conditions. The expression data were normalized using *PPP6R1*, *CCDC97*, *ACTB*, or *EPHA2* RGs individually or the geometric mean of *PPP6R1* + *CCDC97* + *ACTB* or *PPP6R1* + *CCDC97* + *EPHA2*. * *p* = 0.05, ** *p* = 0.01, *** *p* = 0.001.

**Table 1 genes-14-00673-t001:** TMT10 reagents added per condition.

Sample No.	Condition	Reagent
S1	SM1	TMT^10^-126
S2	SM2	TMT^10^-127N
S3	SM3	TMT^10^-127C
S4	OM1	TMT^10^-128N
S5	OM2	TMT^10^-128C
S6	OM3	TMT^10^-129N
S7	AD1	TMT^10^-129C
S8	AD2	TMT^10^-130N
S9	AD3	TMT^10^-130C
Sample Pool 1	SM1–AD3	TMT^10^-131

**Table 2 genes-14-00673-t002:** Ranking of proteins by smallest difference of means between the ADSCs and ADs; the difference between the means for the ADM–SM and ADM–OB conditions are reported, and the *p*-values as determined by ANOVA. MPRA = mean relative protein abundance.

Ranking	Uniprot Accession Number (GN = Corresponding Gene Name)	Protein Description	Difference of Means ADSC-AD (MRPA)	Difference of Means ADSC-OB (MRPA)	Difference of Means AD-OB (MRPA)	ANOVA (*p*-Value) *
1	F6RYC9 GN = *FMC1*	Formation of mitochondrial complex V assembly factor 1 homolog	0.09	203.33	203.25	0.71
2	F6VEQ6 GN = *BCKDK*	Branched chain ketoacid dehydrogenase kinase	1.41	841.22	839.81	0.61
3	F6S466 GN = n.r.	Uncharacterized protein	2.45	362.92	360.47	0.10
4	F7C673 GN = *ARHGAP*12	Rho GTPase activating protein 12	2.66	25.72	28.38	0.99
5	F7DQ27 GN = *GLMN*	Glomulin, FKBP associated protein	2.81	439.74	436.94	0.63
6	F7DY57 GN = *SLC12A5*	Solute carrier family 12 member 5	3.41	152.02	148.61	0.64
7	F6QJT2 GN = *TWSG1*	Twisted gastrulation BMP signaling modulator 1	6.43	122.23	115.80	0.99
8	F7CVS4 GN = *EHD3*	EH domain containing 3	6.89	76.78	83.66	0.91
9	F6XYF8 GN = *TRIM26*	Tripartite motif containing 26	7.04	506.65	499.61	0.46
10	F6SKN5 GN = *LRSAM1*	Leucine rich repeat and sterile α motif containing 1	7.34	77.12	69.78	0.85
11	F6PTR6 GN = *PPP1R9B*	Protein phosphatase 1 regulatory subunit 9B	7.50	3534.30	3526.79	0.37
12	F7D5F3 GN = *SDHC*	Succinate dehydrogenase complex subunit C	7.69	159.14	166.82	0.82
13	F7A2B6 GN = *CHM*	Rab proteins geranylgeranyltransferase component A	7.71	339.83	347.55	0.77
14	F6QAL5 GN = *EPHA2*	EPH receptor A2	8.22	1153.61	1161.84	0.73
15	F6SG13 GN = *MPHOSPH8*	Uncharacterized protein	9.24	51.53	60.77	0.96
16	F6R625 GN = *CCDC97*	Coiled-coil domain containing 97	9.25	159.47	168.72	0.36
17	F6TZB2 GN = *NRBP1*	Nuclear receptor binding protein 1	9.43	64.97	74.40	0.99
18	H9H022 GN = n.r.	PHD finger protein 8	9.46	50.95	60.41	0.70
19	F6V860 GN = *PPP6R1*	Protein phosphatase 6 regulatory subunit 1	9.78	27.80	37.58	0.86
20	F6PMV9 GN = *IRGQ*	Immunity related GTPase Q	10.82	278.48	267.66	0.79
21	F6WE69 GN = *ARHGEF12*	Rho guanine nucleotide exchange factor 12	11.16	86.06	74.90	0.95
22	F6T4G5 GN = *DCAF8*	Uncharacterized protein	12.01	883.61	895.62	0.77
23	F7DNN2 GN = *COQ4*	Ubiquinone biosynthesis protein COQ4 homolog, mitochondrial	12.08	488.65	476.57	0.23

* Proteins demonstrate stable expression levels in all three experimental conditions SM, ADM, and OM. Based on the analysis of variance, *p* > 0.05 indicates no significant difference in relative protein abundance among the three conditions.

**Table 3 genes-14-00673-t003:** Select candidate RGs and TGs, primer sequences, annealing temperatures, amplicon size in base pairs (bp), their respective PCR efficiencies, and mean C_q_ values assessed in a pooled sample consisting of cDNA from parental ADSCs and differentiated adipocytes and osteoblasts, and intron splice site. R^2^: correlation coefficient; E: PCR efficiency (%); STDEV: standard deviation.

Genes (NCBI Record)	Forward and Reverse Primer Sequences (5′ > 3′)	Amplicon Size (bp)	Annealing Temperature for qPCR (°C)	R^2^	E (%)	Mean C_q_ ± STDEV	Intron Splice Site (Nucleotide Position Based on NCBI Record)
*PPP6R1* (XM_014730802.2)	F840 ATTGTCCAGCGGCTCATCGAGC R912 GGGACTGGGACGCATTGGAATG	73	67	0.9535	88.3	24.50 ± 0.2	887, 888
*EHD3* (XM_001918104.5)	F1121 CTTTGGCAATGCCTTCTTGAAC R1217 AGAGAGGATCCCTGGAGTGTCG	97	62	0.9954	107.4	24.51 ± 0.08	1144, 1145
*CCDC97* (XM_001500237.6)	F501 TGGTGACCACCGAGCAGACTTC R647 TGCTCGTCGCTGAAGTACTCGC	147	71	0.9941	109.92	23.67 ± 0.04	622, 623
*EPHA2* (XM_001488739.5)	F764 TGCCCATCGGTCAGTGTCTGTG R888 GTGTGCAGGGCACTCCAAACAG	125	71	0.9842	101.61	20.84 ± 0.14	823, 824
*GAPDH* (NM_001163856.1)	F552 TGGCATCGTGGAGGGACTCATG R642 ATCGCGCCACATCTTCCCAGAG	91	72	0.9783	90.98	17.84 ± 0.23	573, 574
*ACTB* (NM_001081838.1)	F924 CATCGCCGACAGGATGCAGAAG R1060 GCTGGAAGGTGGACAATGAGGC	137	72	0.9329	109.42	16.19 ± 0.35	984, 985
*β2M* (NM_001082502.3)	F14 CTGCTGCTGTGGTAGCTATGGC R118 AAACCTGAACCTTCGGAACACG	105	65	0.9479	83.65	21.19 ± 0.24	97, 98
*FABP5* (XM_001489456.5)	F103 GAAGATGGCGCTTGGTGGAGAG R204 AATCTGGTTTGGCCATTGCACC	102	68	0.9723	103.98	26.75 ± 0.37	156, 157
*RUNX2* (XM_023624251.1)	F847 CTGCTGAGCTCCGAAATGCCTC R942 AACTCTTGCCTCGTCCACTCCG	96	69	0.9348	90.2	23.89 ± 0.14	934, 935

**Table 4 genes-14-00673-t004:** Expression stability ranking of candidate RGs as per BestKeeper, NormFinder, geNorm, ∆C_T_, and RefFinder analysis.

Rank	BestKeeper		NormFinder		geNorm		∆C_T_ analysis		RefFinder Analysis	
	Gene	r * (STDEV, *p*-value)	Gene	Stability Value (SE)	Gene	M Value	Gene	Average of STDEV	Gene	Geometric Mean of Ranking Values
1	*CCDC97*	0.943 (0.29, 0.001)	*PPP6R1*	0.005 (0.007)	*PPP6R1*	0.312	*PPP6R1*	0.42	*PPP6R1*	1.19
2	*PPP6R1*	0.883 (0.19, 0.002)	*CCDC97*	0.005 (0.007)	*CCDC97*	0.329	*CCDC97*	0.46	*CCDC97*	1.86
3	*GAPDH*	0.769 (0.58, 0.015)	*ACTB*	0.010 (0.010)	*EPHA2*	0.354	*ACTB*	0.47	*ACTB*	2.28
4	*EPHA2*	0.463 (0.35, 0.210)	*EPHA2*	0.019 (0.006)	*GAPDH*	0.376	*EPHA2*	0.58	*EPHA2*	4.00
5	*β2M*	0.309 (0.44, 0.418)	*EHD3*	0.026 (0.007)	*ACTB*	0.427	*β2M*	0.62	*β2M*	5.48
6	*ACTB*	0.058 (0.06, 0.885)	*β2M*	0.030 (0.008)	*β2M*	0.526	*GAPDH*	0.66	*GAPDH*	5.96
7	*EHD3*	−0.057 (0.44, 0.885)	*GAPDH*	0.040 (0.010)	*EHD3*	0.601	*EHD3*	0.68	*EHD3*	6.44

* r = Pearson correlation coefficient; STDEV = standard deviation; SE = standard error.

## Data Availability

Not applicable.

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
