# Peer review of "Selection and Validation of Reference Genes for Gene Expression Studies in an Equine Adipose-Derived Mesenchymal Stem Cell Differentiation Model by Proteome Analysis and Reverse-Transcriptase Quantitative Real-Time PCR"

_genes, 2023, doi:10.3390/genes14030673_

Round 1
Reviewer 1 Report
This work describes substantial effort in identifying stable reference genes in an equine adipose-derived mesenchymal stem cell differentiation model followed by validating SYBR real-time qPCR assays for these reference genes. Proteomic data was used to identify proteins with consistent expression across cell types. I appreciate the amount of effort invested in validating a robust set of reference gene assays, particularly in using multiple algorithms to evaluate candidate reference genes as well as the very thorough methods section. However, there are several discrepancies that should be addressed prior to publication.
Comments:
1) While producing replicate cultures is commendable for validation efforts, I was surprised that the same cell culture wasn’t split for RNA, DNA, and proteome studies. Using different cultures for nucleic acid and protein studies allows for inconsistencies between the datasets. Is there a rationale for using different cultures?
2) It is great to see the efficiency and R2 values reported for the assays but several values are considered unacceptable for real-time PCR assays because of low efficiency (less than 90%) or low R2 values (<0.98) (these are widely accepted parameters, cited by Johnson et al in Real-time quantitative PCR, pathogen detection and MIQE, 2013, DOI: 10.1007/978-1-60327-353-4_1; also https://www.thermofisher.com/us/en/home/life-science/pcr/real-time-pcr/real-time-pcr-learning-center/real-time-pcr-basics/real-time-pcr-troubleshooting-tool/gene-expression-quantitation-troubleshooting/poor-pcr-efficiency.html#:~:text=The%20efficiency%20of%20the%20PCR%20should%20be%20between%2090%E2%80%93100,the%20PCR%20has%20poor%20efficiency). With the review of RT-qPCR assay design and reference genes provided in the introduction, it is surprising that this was not addressed by the authors during assay optimization. Further, some assays use unusually high annealing temperatures (such as 67°C), which could affect amplification efficiency. Please explain why these assays were not optimized to improve efficiency and R2 values when such huge efforts went into the earlier stages of this study. This could impact the relative expression results as some assays with the lowest efficiency and R2 values were used as reference genes such as PPP6R1 (both low: 88% and 0.95) and ACTB (efficiency on the upper border at 109.4 but R2 low at 0.93). Please address the use of assays with low performance metrics.
There is no R2 value reported for the RUNX2 assay in what should be Table 3 on page 14.
3) The mRNA relative expression levels didn’t reflect the protein levels as nicely as expected but there can be reasons for that. In fact, the plots in Figure 4 look like the labels could have been switched. While FABP5 expression was significantly increased in AD cells at both mRNA and protein levels, FABP5 mRNA expression was also significantly increased in OB cells. Please discuss whether this finding was expected and add possible explanations.
4) On lines 566-7, the authors state that RUNX2 mRNA levels were “slightly higher” in OBs than ADSCs but Figure 4 indicates that there is no significant difference. Please clarify this discrepancy.
5) Line 567 stated that RUNX2 protein levels were not significantly higher in OBs compared to ADSC, but Figure 2 indicates that the difference was significant (p<0.001). Please clarify this discrepancy.
Minor comments
-Revise reference numbers, number 16 precedes 15 on lines 375 and 376
-Remove comma on line 209 after “with”
-Replace “flaked” with “flanked” on line 453
-Table numbering incorrect starting with what should be Table 2 on page 12 (currently labeled Table 1)
Author Response
Thank you very much for thoroughly reviewing our paper, your comments were most helpful. Amendments were made as described:
1) While producing replicate cultures is commendable for validation efforts, I was surprised that the same cell culture wasn’t split for RNA, DNA, and proteome studies. Using different cultures for nucleic acid and protein studies allows for inconsistencies between the datasets. Is there a rationale for using different cultures?
A single batch of adipose-derived mesenchymal stem cells were prepared and expanded to the same passage for all studies, we were limited to an extent by the cell numbers as we did not want to have high passage cells. Cells were split simultaneously into separate 6 well plates or T75 flasks, so that sufficient material would be available to conduct each analysis. Since different buffers were required for protein vs nucleic acid extraction, these cells were maintained in different wells to ease processing. As mentioned in your review, we ran three biological replicates for each condition to ensure that the differentiation processes were standardized. This was also done for the Oil Red O and Von Kossa-stained controls.
2) It is great to see the efficiency and R2 values reported for the assays but several values are considered unacceptable for real-time PCR assays because of low efficiency (less than 90%) or low R2 values (<0.98) (these are widely accepted parameters, cited by Johnson et al in Real-time quantitative PCR, pathogen detection and MIQE, 2013, DOI: 10.1007/978-1-60327-353-4_1; also https://www.thermofisher.com/us/en/home/life-science/pcr/real-time-pcr/real-time-pcr-learning-center/real-time-pcr-basics/real-time-pcr-troubleshooting-tool/gene-expression-quantitation-troubleshooting/poor-pcr-efficiency.html#:~:text=The%20efficiency%20of%20the%20PCR%20should%20be%20between%2090%E2%80%93100,the%20PCR%20has%20poor%20efficiency). With the review of RT-qPCR assay design and reference genes provided in the introduction, it is surprising that this was not addressed by the authors during assay optimization. Further, some assays use unusually high annealing temperatures (such as 67°C), which could affect amplification efficiency. Please explain why these assays were not optimized to improve efficiency and R2 values when such huge efforts went into the earlier stages of this study. This could impact the relative expression results as some assays with the lowest efficiency and R2 values were used as reference genes such as PPP6R1 (both low: 88% and 0.95) and ACTB (efficiency on the upper border at 109.4 but R2 low at 0.93). Please address the use of assays with low performance metrics.
In the case of primer design, we wanted to be sure that the primers flanked intron splice sites to control for genomic DNA contamination. This limited us to the location for primer placement. Hence in the case of PPP6R1 primers, these were the most efficient we could design; other sets were trialed (data not shown). We also attempted to design primers for other reference gene targets, but primer efficiencies were lower especially for primers that flanked the intron splice sites. Another challenge we faced was that the annotated equine gene sequences (details on intron splice sites were not available) for many of the equine reference gene candidates identified by proteomics were not available so we were not able to design primers for those possible reference gene candidates. These comments were added as a study limitation under section 4.3.
With respect to the annealing temperatures, the Tm for primer design was set to => 57 °C to assure specificity. Also a temperature gradient (60 °C- 75 °C) was run for all primer sets selected to determined the annealing temperature at which primer amplification was most optimal (The temp. at which the most cDNA product was produced in a set number of amplification cycles with no non-specific amplification was considered optimal). In the case of the PPP671 primer set, 67 °C was found to be most optimal temperature for amplification.
2b) There is no R2 value reported for the RUNX2 assay in what should be Table 3 on page 14.
The R2 value was accidentally deleted as the table was being formatted. It has been added to the table and is 0.9348.
3) The mRNA relative expression levels didn’t reflect the protein levels as nicely as expected but there can be reasons for that. In fact, the plots in Figure 4 look like the labels could have been switched. While FABP5 expression was significantly increased in AD cells at both mRNA and protein levels, FABP5 mRNA expression was also significantly increased in OB cells. Please discuss whether this finding was expected and add possible explanations.
I did double check the data and the plots are labelled accurately. We did not expect the RNA and protein levels to match always, since gene expression can be either post-transcriptionally or post-translationally regulated. For this reason we made a longer list of possible candidates based on the proteomic data than we thought we would need. It is possible that FABP5 expression is post-translationally regulated in osteoblasts so that a protein degradation method such as ubiquitination, for example, could be targeting FABP5 protein for destruction, keeping protein levels low, irrespective of mRNA levels. At present there are no publication in the literature to describe how Fabp5 protein expression levels are regulated in equine ADSCs, ADs or OBs.
4) On lines 566-7, the authors state that RUNX2 mRNA levels were “slightly higher” in OBs than ADSCs but Figure 4 indicates that there is no significant difference. Please clarify this discrepancy.
This discrepancy was corrected as there is no significant difference for RUNX2 mRNA expression levels in OBs versus ADSCs.
5) Line 567 stated that RUNX2 protein levels were not significantly higher in OBs compared to ADSC, but Figure 2 indicates that the difference was significant (p<0.001). Please clarify this discrepancy.
This error was corrected, there was no significant difference at the mRNA level, but there was a significant difference at the protein level as described in the discussion section.
Minor comments
-Revise reference numbers, number 16 precedes 15 on lines 375 and 376
References are revised.
-Remove comma on line 209 after “with”
Comma removed.
-Replace “flaked” with “flanked” on line 453
Corrected.
-Table numbering incorrect starting with what should be Table 2 on page 12 (currently labeled Table 1)
Table numbers were corrected.

Reviewer 2 Report
In their study, Riveroll et al identified novel reference genes in adipose-derived stem cells (ADSCs) and differentiated cell types, adipocytes (ADs) and osteoblasts (OBs) to facilitate investigations into gene expression changes during equine adipose-derived mesenchymal stem cell differentiation. The authors also conducted a repeat study and refuted previously published findings that GAPDH, β-actin, and β2-microglobulin were not stably expressed in adipose-derived mesenchymal stem cells, while providing explanations for the discrepancies between their results and those of other laboratories. In summary, the authors' results are trustworthy and their conclusions are well-defined, although minor revisions are necessary, such as correcting mislabeled table numbers.
Author Response
Thank you for thoroughly reviewing our paper, your revisions were helpful and the paper was amended as follows:
Main Point:
Minor revisions are necessary, such as correcting mislabeled table numbers.
The paper was reviewed thoroughly and mislabeled table numbers were corrected.

Reviewer 3 Report
The manuscript by Riveroll et al applied MS and RT-qPCR to select the reliable RGs for Adipose-derived MSC differentiation. The finding of reliable RGs could be of use to the scientific community who investigate the Adipose-derived MSC differentiation. Overall, the description of the manuscript was well written. The results and conclusions are clearly presented. However, there are some points that could be addressed to enhance the quality of the manuscript.
1. The Figure quality was low. High-resolution of the figures should be provided.
2. Western blot of the selected and commonly RGs should be provided.
3. By using the reliable RGs, what it's the new TGs/pathways identified that could be used to indicate the successful differentiation of the Adipose-derived MSCs. This part should be more developed.
Author Response
Thank you very much for taking the time to carefully review our paper, it is most appreciated. Your points were thoroughly considered and amendments made to this paper as described below. We also take into considerations your comments for a follow on paper.
- The Figure quality was low. High-resolution of the figures should be provided.
We will check with journal editors as to the quality specifications of figures required and will revise accordingly.
- Western blot of the selected and commonly RGs should be provided.
Proteomic data was provided instead to demonstrate protein identity and expression levels. Since proteomic data is based on unique protein fingerprints generated by trypsin digestion at arginine and lysine residues, protein identification is considered reliable. Furthermore, the relative abundance abundances of the proteins from condition to condition can be compared, since samples were each labelled with a unique isobaric tag (Table1) that allowed for data sorting after protein identification by LSMS3. Therefore, the measured differences in protein expression levels for the proposed RGs for the two test conditions (AD and OB) could each be compared to the untreated condition (SM) and reported as fold-change. Also, experiments were done in triplicate and there was no significant difference in protein expression among the reference genes between the different conditions (Figure 3).
- By using the reliable RGs, what it's the new TGs/pathways identified that could be used to indicate the successful differentiation of the Adipose-derived MSCs. This part should be more developed.
We agree that we can use the full proteomic data set to describe in more detail the differences in protein expression profiles among progenitor mesenchymal stem cells, adipocytes and osteoblasts. In this first paper, we rely on oil Red O staining to show an increase in lipid production and also a relative increase in the expression of Fatty Acid Binding Protein 5 (Fabp5) in adipocytes, both markers of adipocyte differentiation, based on the literature. We plan to fully describe the differences in protein expression among progenitor stem cells, adipocytes and osteoblasts in a follow on paper. For example FABP5 is reported to be a positive regulator of PPARγ, (Yu et al 2016) a well-known regulator of adipogenesis. Also FABP5 is reported to be a transcriptional and metabolic regulatory factor that is secreted by adipocytes (Yamamoto et al 2016). We have added the Yamamoto findings and reference to the paper in an effort to further support FABP5 as a marker for adipocytes and its involvement in adipocyte function.
Yu, C. W., Liang, X., Lipsky, S., Karaaslan, C., Kozakewich, H., Hotamisligil, G. S., et al. (2016). Dual Role of Fatty Acid-Binding Protein 5 on Endothelial Cell Fate: a Potential Link between Lipid Metabolism and Angiogenic Responses. Angiogenesis 19 (1), 95–106. doi:10.1007/s10456-015-9491-4
Yamamoto T, Furuhashi M, Sugaya T, Oikawa T, Matsumoto M, Funahashi Y, Matsukawa Y, Gotoh M, Miura T. Transcriptome and Metabolome Analyses in Exogenous FABP4- and FABP5-Treated Adipose-Derived Stem Cells. PLoS One. 2016 Dec 9;11(12):e0167825

Round 2
Reviewer 3 Report
The authors have made significant improvements to the manuscript and address all our concerns. I would be happy to recommend publication in Genes.